# Long-Term Surgical Results of Skip Pedicle Screw Fixation for Patients with Adolescent Idiopathic Scoliosis: A Minimum-Ten-Year Follow-Up Study

**DOI:** 10.3390/jcm9124002

**Published:** 2020-12-10

**Authors:** Masashi Uehara, Shugo Kuraishi, Shota Ikegami, Hiroki Oba, Takashi Takizawa, Ryo Munakata, Terue Hatakenaka, Michihiko Koseki, Jun Takahashi

**Affiliations:** 1Department of Orthopaedic Surgery, Shinshu University School of Medicine, 3-1-1 Asahi, Matsumoto, Nagano 390-8621, Japan; kura@apricot.ocn.ne.jp (S.K.); sh.ikegami@gmail.com (S.I.); oba1@hotmail.co.jp (H.O.); takizawahal449@gmail.com (T.T.); battlemiddle3@yahoo.co.jp (R.M.); hatakenaka@shinshu-u.ac.jp (T.H.); jtaka@shinshu-u.ac.jp (J.T.); 2Department of Mechanical Engineering and Robotics, Faculty of Textile Science and Technology, Shinshu University, 3-15-1 Tokida, Ueda, Nagano 390-8621, Japan; koseki@shinshu-u.ac.jp

**Keywords:** long-term results, adolescent idiopathic scoliosis, skip pedicle fixation, 10 years, posterior fusion, surgery

## Abstract

Skip pedicle screw fixation for adolescent idiopathic scoliosis (AIS) requires fewer screws and can reduce the risk of neurovascular injury as compared with segmental pedicle screw fixation. However, the long-term impact of screw number reduction on correction and clinical results is unclear. This study examined the 10-year post-operative outcomes of skip pedicle screw fixation for patients with AIS. We reviewed the outcomes of 30 patients who underwent skip pedicle screw fixation for AIS. Radiological and clinical findings were assessed before and immediately, 2 years, and 10 years after surgery in the remaining 25 patients. The mean Cobb angle of the main curve preoperatively and immediately, 2 years, and 10 years post-operatively was 59.4°, 23.4°, 25.8°, and 25.60°, respectively, and was significantly improved at all post-surgical time points (all *p* < 0.001). The mean correction rate immediately after surgery was 60.8%, and the correction loss rate at the observation end point was 4.8%. The Cobb angle of the lumbar curve was significantly improved immediately after surgery, and the correction persisted until 10 years post-operatively. Remarkable gains were observed for most Scoliosis Research Society-22 patient questionnaire sub-scores at the final follow-up versus preoperative assessments. In conclusion, good correction of the AIS deformity by skip pedicle screw fixation was well maintained over a long follow-up period of 10 years, with clinically meaningful gains in Society-22 patient questionnaire sub-scores.

## 1. Introduction

Segmental pedicle screw fixation using numerous thoracic pedicle screws as described by Suk et al. [1] is a useful option for posterior spinal fusion in adolescent idiopathic scoliosis (AIS). The posterior approach offers stronger fixation strength to enable shorter fusion length and better correction [2]. On the other hand, pedicle screw fixation has also been associated with a risk of injury to neurovascular and visceral structures, such as the spinal cord, nerve root, lung, and aorta [1,3,4,5]. To avoid those serious complications, Takahashi et al., employ skip pedicle screw fixation for the AIS deformity since it requires fewer screws (Figure 1) and can reduce surgical costs [6,7,8,9]. Uehara et al., earlier described the two-year results of computer-assisted skip pedicle screw fixation in 62 consecutive cases of AIS containing all Lenke classification types apart from 5C [9]. However, evidence on the long-term outcomes of skip pedicle screw fixation, especially scoliosis correction loss, implant failure including screw breakage/pull-out, and decreased Scoliosis Research Society-22 patient questionnaire (SRS-22r) scores, is scarce.

The purpose of this study is to examine if the corrective position retention and good clinical outcomes of skip pedicle screw fixation are maintained even at 10 years after surgery. To further validate the usefulness of skip pedicle screw fixation for AIS, we examined the radiological and clinical findings of patients after a long observation period of at least 10 years.

## 2. Methods

### 2.1. Patients

This study was conducted at our institution. We earlier described the 2-year results of skip pedicle screw fixation in 62 consecutive cases of AIS [9]. Among 30 consecutive patients receiving preoperative computed tomography (CT)-based navigation-assisted skip pedicle screw fixation for AIS at our hospital between August 2005 and March 2010, 25 patients (2 male and 23 female; mean ± standard deviation (SD) age: 14.4 ± 2.1 years, range: 12–19 years) who were followed for a minimum of 10 years (follow-up rate: 83.3%) were retrospectively reviewed. All surgeries were performed by one senior spine surgeon specializing in scoliosis (J.T.). The inclusion criteria were patients who underwent AIS surgery and follow-up at our hospital. The exclusion criteria were follow-up of less than 10 years and Lenke type 5 curves. Five patients without follow-up of at least 10 years due to relocation were excluded. Classification according to Lenke grading was type 1A in seven patients, 1B, 2A, and 6C in three patients each, 1C in four patients, 4B in one patient, 3C and 4C in two patients each. Patients with Lenke type 5 curves were excluded from this investigation since they often received segmental pedicle screw fixation. Prior approval of the study was obtained from the investigational review board of our hospital (No. 3500). Written informed consent for publication was obtained from all patients prior to this study. No financial support or equivalent was received for this investigation. The complete database of the cohort can be accessed at the Zenodo repository (doi.org/10.5281/zenodo.4295859).

### 2.2. Surgical Technique

Pedicle screw insertion into the upper and lower ends of the instrumentation area was performed bilaterally. Screw placement in the vertebral levels other than the upper instrumented vertebra, lowest instrumented vertebra, and apex was judged based on the size and rigidity of the curve and was skipped when possible. Screw insertion was skipped near the apical vertebrae of each curve and in the junctional zone between structural and compensatory curves that were not close to fused end vertebrae. We excluded pedicles with an outer diameter of less than the thinnest screw diameter during navigation planning. Pedicle screws were routinely inserted near the concave apical vertebrae. For cases in which a 4.5 mm-diameter pedicle screw appeared difficult to insert in navigation planning, tape was used instead. However, in cases where only one vertebra was skipped, no tape was used. Since pedicle diameter is wider near the convex vertebrae, we basically used pedicle screws and skipped one vertebra per inserted vertebra. Screw holes were made using a Kotani probe [10] with CT-based navigation, while pedicle screws were inserted without navigation. Bone grafting employed local bone. Ponte osteotomy was not performed in this cohort. The lower instrumented vertebra (LIV) was decided as the vertebra which last touched the central sacral vertical line when the lumbar modifier was Lenke 1A or 2A. For Lenke 1B and 1C curves, the LIV was determined as the stable vertebra or one vertebra distal to the stable vertebra. For Lenke 3C, 4B, 4C, and 6C curves, we routinely set the LIV as L3.

### 2.3. Evaluation and Statistical Analysis

The first author (M.U.), who was not associated with any surgery in the cohort, performed the evaluation. Cobb angle [11] and thoracic kyphosis from T5 to T12, absolute value of clavicular angle (CA), distance from the C7 plumb line to the central sacral vertical line (C7PL), LIV tilt, and other radiological parameters were evaluated before and immediately, 2 years, and 10 years after surgery. The main curve flexibility was calculated as: (Cobb angle in standing position—Cobb angle in side bending)/Cobb angle in standing position × 100%. CT was performed at 6 months postoperatively to confirm bony fusion of the facet and to determine whether the patients could be permitted to exercise. Clinical outcomes were examined using SRS-22r scores [12,13] before and at 2 and 10 years post-operatively. Results for the Japanese Orthopaedic Association Back Pain Evaluation Questionnaire (JOABPEQ), which included a visual analog scale (VAS) for back pain and complications requiring surgical revision, were assessed at the study end point. The JOABPEQ contains five domains (pain-related disorder, lumbar spine dysfunction, gait disturbance, social life disturbance, and psychological disorder) whose scores range from 0 (lowest) to 100 (highest) [14]. We also evaluated whether JOABPEQ scores for back pain were influenced by poor overall correction, such as coronal decompensation, lumbar decompensation, the adding-on phenomenon, and early degeneration. Coronal decompensation was considered as C7PL deviation exceeding 20 mm [15]. Lumbar decompensation was defined as progression of thoracolumbar/lumbar Cobb angle by 10° or more versus immediately postoperative Cobb angle [16]. The adding-on phenomenon was judged as follows: (1) a ≥5° aggravation of scoliotic Cobb angle and a shift in curve-end vertebrae towards the caudal side, or (2) a ≥5° wedge-shaped change in the vertebral disc adjacent to the lower side of the fused vertebrae during postoperative follow-up [17]. Pre- and post-surgical data were compared using the paired t-test with Bonferroni correction, and data from the two groups were compared by means of Welch’s t-test with EZR software (Saitama Medical Center, Jichi Medical University, Saitama, Japan), a graphical user interface for R (The Foundation for Statistical Computing, Vienna, Austria). As a modified version of R commander, EZR adds statistical functions frequently used in biostatistics. A *p*-value of <0.05 was considered statistically significant.

## 3. Results

The cohort’s preoperative radiological and clinical features are summarized in Table 1. The preoperative Cobb angle of the main thoracic curve was 59.4°, and the main thoracic curve flexibility was 38.2%. The mean ± SD number of fused vertebrae was 10.8 ± 1.9 (range: 8–13), mean surgical time was 294 ± 80 min (range: 168–420 min), mean blood loss volume was 1304 ± 672 mL (range: 500–3050 mL), and mean screw density (i.e., number of screws per vertebra fused) was 1.33 ± 0.22 (range: 1.00–1.77). The vertebral level of the LIV was T12 in five cases, L1 in eight cases, L2 in three cases, and L3 in nine cases. Among the 250 inserted pedicle screws, the grade 2 and 3 perforation (i.e., >2 mm) rate was 6.8%, and the grade 3 perforation (i.e., >4 mm) rate was 4.4% based on post-operative CT. No severe neurovascular injuries from screw misplacement were observed.

The Cobb angle of the main thoracic curve was significantly improved immediately, at 2 years, and at 10 years after correction vs. preoperative measurements (all *p* < 0.001) (Figure 2, Table 2). The mean correction rate of the main curve at the above time points was 60.8 ± 2.2%, 56.3 ± 2.5%, and 56.8 ± 2.2%, respectively (Table 2). The correction loss rate between immediately and 10 years after surgery was 4.8 ± 3.6%.

The T5–12 thoracic kyphotic angle was significantly improved at all post-operative time points over baseline (all *p* < 0.05) (Figure 2, Table 2). Absolute CA values were similar, corresponding to 2.2 ± 0.4° preoperatively, 3.2 ± 0.5° immediately after surgery, 2.4 ± 0.4° at 2 years after surgery, and 2.2 ± 0.4° at the final follow-up. Similarly, absolute C7PL values remained comparable, with values of 1.4 ± 0.2 cm, 1.3 ± 0.2 cm, 0.7 ± 0.2 cm, and 1.4 ± 0.4 cm, respectively (Table 2).

The Cobb angle of the lumbar curve and LIV tilt were significantly improved immediately after surgery, which persisted until 10 years post-operatively (Figure 2, Table 2).

SRS-22r scores increased significantly in all domains from preoperatively to 2 and 10 years after surgery (all *p* < 0.05) (Table 3). At the study end point, the mean satisfaction score was 4.02 ± 0.17 (range: 2–5.0), and the mean total score was 4.29 ± 0.09 (range: 3.1–4.95) (Table 3).

JOABPEQ scores at 10 years after surgery were 86.8 ± 4.2 for pain-related disorder, 94.0 ± 2.4 for lumbar spine dysfunction, 95.8 ± 2.7 for gait disturbance, 84.4 ± 3.6 for social life disturbance, and 72.9 ± 3.4 for psychological disorder (Table 4). Three (12.0%) patients experienced coronal decompensation, five (20.0%) patients had lumbar decompensation, two (8.0%) patients exhibited adding-on, and three (12.0%) patients had early degeneration. The JOABPEQ score for low back pain in patients with coronal decompensation was 100 ± 0, which was significantly higher than in patients without it (85.0 ± 21.8) (*p* = 0.004). The JOABPEQ scores for low back pain in patients with and without lumbar decompensation were comparable, with values of 91.4 ± 19.2 and 85.7 ± 21.8, respectively (*p* = 0.58). The JOABPEQ score for low back pain in patients with adding-on was significantly higher than in patients without it (100 ± 0.0 vs. 85.7 ± 21.6, *p* = 0.004). The respective JOABPEQ scores for low back pain in patients with and without early degeneration were similar, corresponding to 62.0 ± 32.9 and 90.2 ± 17.4 (*p* = 0.27). The mean VAS score for low back pain was 19.6 ± 5.4 (range: 0−80) and was zero in 13 (52%) patients. Bone union was confirmed by CT in all cases. Screw breakage without symptoms requiring surgical revision was seen in two patients. No severe neurovascular complications from surgical invasion or indication for surgical revision were recorded.

## 4. Discussion

Skip pedicle screw fixation for AIS correction significantly improved both radiological and clinical parameters at a minimum of 10 years after surgery using fewer inserted screws than in segmental fixation and without complications requiring surgical revision.

The primary goals of surgical AIS treatment are preventing curve progression and correcting the deformity. Segmental pedicle screw fixation is frequently used for rod anchoring in posterior fixation of the scoliotic deformity in AIS. The rate of neurovascular complications from misplaced screws ranges from 0% to 1.3%, being attributed mainly to narrow pedicle diameter and spinal rotation [2,18,19,20]. We have been employing skip pedicle screw fixation for AIS to reduce the number of screw violations and the associated risk of major adverse events [6]. Behrbalk et al., reported that a low screw density technique was as safe and effective as a high screw density technique for posterior-only correction of Scheuermann kyphosis and could provide significant cost savings [21]. Cheung et al., reported that fulcrum bending correction could estimate curve correction in AIS surgery using alternate-level thoracic pedicle screws [22]. In our study, the preoperative Cobb angle of the main curve was 59.4°, and the main curve flexibility was 38.2%. We defined the planned correction angle as the difference between the Cobb angle in side bending and the target Cobb angle, and the number of screws was determined with reference to a previously reported formula, i.e., planned correction angle/1.7° [23]. In the present study, mean screw density was 1.33, grade 3 screw perforation rate was 4.4%, and grade 2 and 3 combined perforation rate was 6.8%. No neurovascular injuries were encountered during 10 years of observation. Although preoperative CT-based navigation is useful for safely inserting pedicle screws, performing a CT scan of all vertebrae for the length of the fusion to check pedicle anatomy and size carries the potential risk of long-term damage from radiation.

Although skip pedicle screw fixation is considered safer than segmental screw fixation, a trade-off with diminished long-term correction and clinical effects may exist. A two-year observational radiological study of surgical AIS correction showed that an all-pedicle screw system provided better maintenance of corrective parameters than did hybrid instrumentation surgery using pedicle screws, hooks, and sublaminar wire or tape as an anchor, with final correction rates of 70.41% and 60.00%, respectively [24]. Preoperative flexibility was approximately 50% in both groups [24]. Hwang et al., also reported satisfactory results for skip pedicle screw insertion, obtaining 69% for correction rate and 2% for correction loss rate at 5 years [25]. In this study, the correction rate was 59.8% immediately post-operatively, and the 10-year correction loss rate was 4.8%, which were less favorable than Hwang’s findings but comparable to those of the hybrid group. Moreover, the correction loss at 10 years was low at 2.2°. The absence of Ponte osteotomy procedures in this series may have reduced the correction rate. Screw breakage was seen in two patients, who experienced a mean correction loss of 10.5° at 10 years but maintained good SRS-22r scores. Regarding the changes occurring from 2 years to 10 years post-operatively, the end vertebrae were not fused in some cases, so it was conceivable that alterations were caused by angular deformations in the cranial or caudal adjacent intervertebrae of the unfused end vertebrae.

Ishikawa et al., reported that the final Cobb angle of the thoracolumbar/lumbar curve was significantly correlated with the immediately post-operative LIV tilt [26]. On the other hand, Skaggs et al., described that LIV tilt was not associated with post-surgical lumbar Cobb angle [27]. In our study, the Cobb angle of the lumbar curve and the LIV tilt were significantly improved until 10 years after surgery. However, 12.0% of the patients had coronal decompensation, 20.0% of the patients had lumbar decompensation, 8.0% of the patients had adding-on, and 12.0% of the patients had early degeneration. Furthermore, comparisons of the LIV tilt immediately after surgery in patients with and without lumbar decompensation revealed similar findings (6.4 ± 5.3° and 8.2 ± 7.3°, respectively; *p* = 0.63).

The JOABPEQ domain scores in our cohort were all considerably higher than the reference values of low-back-pain patients [28] (Table 4). Furthermore, all scores were greater than the minimum clinically important differences (MCIDs) for JOABPEQ (20.4 for low back pain, 15.6 for lumbar function, 16.8 for walking ability, 13.4 for social life function, and 9.4 for mental health) reported by Ogura et al. [29]. The score for psychological disorder was the lowest, possibly due to the stresses of child care or working life in our young adult cohort. Hence, the long-term correction ability of skip pedicle screw fixation was considered sufficient, especially since the preoperative flexibility in our cohort (approximately 40%) was comparably lower than that of earlier reports. Furthermore, evaluations on whether the JOABPEQ scores for back pain were influenced by poor overall correction, such as coronal decompensation, lumbar decompensation, the adding-on phenomenon, and early degeneration, revealed that coronal decompensation and adding-on were not relevant adverse effects at 10 years after surgery.

Regarding long-term radiological results, the main thoracic curve Cobb angle improved from 51° to 16°, with a correction loss rate of 5% in a five-year study on segmental pedicle screw fixation for thoracic scoliosis [30]. However, data on the long-term benefits of skip pedicle screw fixation are scarce. We observed that this method significantly improved and maintained the coronal and sagittal radiological parameters of AIS cases at 10 years after surgery without severe neurovascular events or the need for surgical revision. Although a formal cost-effectiveness analysis was not performed, fewer implants were presumed as more economical for the patients.

With regard to the long-term clinical results of surgical AIS treatment, there are several reports on the Harrington method, posterior fixation with the rod and hook system, and anterior fusion, and few on pedicle screw fixation. In a comparative control study using the SRS-24, AIS patients undergoing correction surgery by the Harrington method had findings equivalent to those of healthy subjects for all parameters, and the magnitude of the scoliotic angle was not associated with quality of life (QOL) scores [31]. In a similar study with healthy subjects using the SRS-22, however, Akazawa et al., witnessed that patients had comparable scores for pain and mental health, but inferior results for function and self-image [32]. Using the Short Form 36 Health Survey, Götze et al., observed that adolescents who received correction surgery by the Harrington method had similar physical function, but lower mental health, in relation to German national standard values. Curve type and magnitude of the scoliotic angle were unrelated to QOL [33]. The present study evaluated 10-year post-operative clinical results using SRS-22r scores to reveal that self-image and sub-total were significantly ameliorated at 2 years and 10 years after surgery. Although some patients were subjected to long fusion procedures, the function and pain scores were also improved after correction. Thus, skip pedicle screw fixation for AIS produces good long-term correction without function loss or pain worsening.

We lastly examined whether the SRS-22r score changes in our cohort were clinically meaningful. In earlier studies on MCID for clinical domain outcomes, MCID was 0.20 for pain (area under the receiver operating characteristic curve (AUC) = 0.723), 0.08 for function (AUC = 0.648), and 0.98 for self-image (AUC = 0.629) [34]. The changes in the scores for pain, function, and self-image were all higher than their respective MCID, confirming that skip pedicle fixation produced clinically meaningful improvements.

Newton et al., noted that posterior segmental instrumentation for AIS could significantly decrease thoracic kyphosis [35]. Such a finding was absent in our study, with the mean thoracic kyphotic angle before and at 2 and 10 years after surgery being 9.1°, 15.2°, and 17.9°, respectively. Skip pedicle screw fixation may also have the advantage of improving thoracic kyphosis. If more rigid cobalt chromium rods had been used, larger thoracic kyphosis might have been possible.

The main limitations of this study were no control group, a limited sample size, and a retrospective design. No blinding of the assessor was also a limitation of this study. Although five patients were lost to follow-up, the resulting 83.3% follow-up rate was considered satisfactory based on Solberg et al., reporting that a follow-up loss of 22% did not bias conclusions about the effects of treatment [36]. Thus, our analysis on the outcomes of 25 AIS patients indicated good long-term radiological and clinical results for skip pedicle screw fixation, with almost all parameters being well conserved up to 10 years post-operatively.

## 5. Conclusions

Skip pedicle screw fixation for AIS provided significant and sustained radiological and clinical improvements at 10 years after surgery without serious complications.

## Figures and Tables

**Figure 1 jcm-09-04002-f001:**
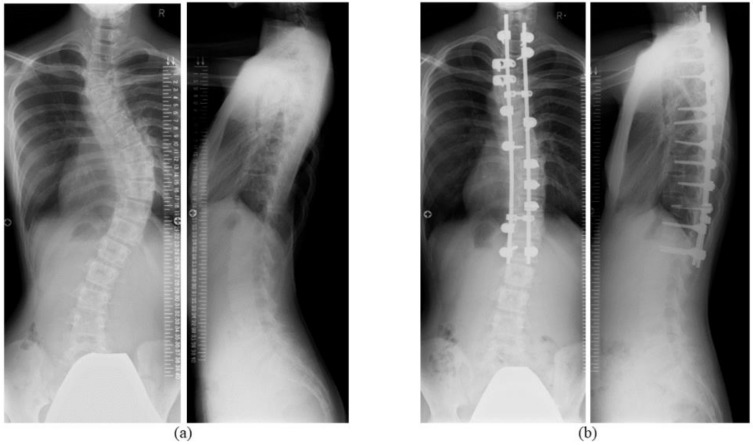
Representative skip pedicle screw fixation for adolescent idiopathic scoliosis. (**a**) Preoperative Cobb angle of the main thoracic curve was 53°. (**b**) Skip pedicle screw fixation from T1 to L1 improved the scoliotic curve to 17°.

**Figure 2 jcm-09-04002-f002:**
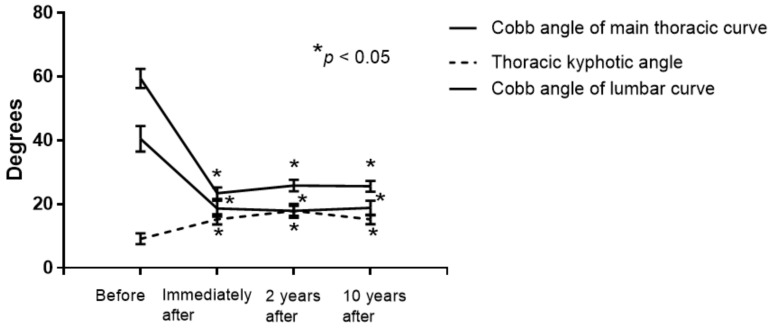
The mean Cobb angle of the main thoracic curve before and at 2 and 10 years after surgery was 59.4°, 25.8°, and 25.6°, respectively. The main thoracic curve Cobb angle was significantly improved at both post-operative time points (*p* < 0.001). The respective mean T5–12 thoracic kyphotic angle before and at 2 and 10 years after surgery was 9.1°, 17.9°, and 15.2°. The Cobb angle of the lumbar curve was significantly improved immediately after surgery, which persisted until 10 years post-operatively.

**Table 1 jcm-09-04002-t001:** Preoperative radiological and clinical features.

	Mean ± Standard Deviation (Range)
Age, years	14.4 ± 2.1 (12–19)
Sex, male/female	2:23
Cobb angle of main thoracic curve, °	59.5 ± 15.1 (44–94)
Main curve flexibility, %	38.2 ± 19.4 (6.3–96.1)
Cobb angle of lumbar curve, °	40.5 ± 19.9 (12–82)
Thoracic kyphotic angle (T5–T12), °	9.1 ± 8.3 (−8–29)
Clavicular angle, °	2.2 ± 1.9 (0–7.9)
C7PL, cm	1.4 ± 1.1 (0–5.0)

C7PL: distance from the C7 plumb line to the central sacral vertical line.

**Table 2 jcm-09-04002-t002:** Surgical results of skip pedicle screw fixation.

Radiological and Physical Evaluation	Before Surgery	Immediately after Surgery	Two Years after Surgery	Ten Years after Surgery
Cobb angle of main thoracic curve, °	59.4 ± 3.0	23.4 ± 1.8 **	25.8 ± 1.7 **	25.6 ± 1.7 **
Correction rate of Cobb angle (main thoracic curve), %	N/A	60.8 ± 2.2	56.3 ± 2.5	56.8 ± 2.2
Thoracic kyphotic angle (T5–T12), °	9.1 ± 1.7	15.2 ± 1.6 **	17.9 ± 1.5 **	15.2 ± 1.5 **
Clavicular angle, °	2.2 ± 0.4	3.2 ± 0.5	2.4 ± 0.4	2.2 ± 0.4
C7PL, cm	1.4 ± 0.2	1.3 ± 0.2	0.7 ± 0.2 *	1.4 ± 0.4
Cobb angle of lumbar curve, °	40.5 ± 4.0	18.6 ± 2.5 **	17.9 ± 2.2 **	18.8 ± 2.3 **
LIV tilt, °	20.1 ± 1.5	7.8 ± 1.4 **	7.3 ± 1.4 **	7.7 ± 1.2 **

All values are expressed as the mean ± standard error; LIV: lower instrumented vertebra, N/A: not applicable; * *p* < 0.05 vs. before surgery, ** *p* < 0.001 vs. before surgery.

**Table 3 jcm-09-04002-t003:** Clinical results of SRS-22r scores.

SRS-22r Domain	Before Surgery	Two Years after Surgery	Ten Years after Surgery
Function	4.31 ± 0.12	4.74 ± 0.05 **	4.65 ± 0.09 *
Pain	4.01 ± 0.12	4.68 ± 0.07 **	4.40 ± 0.13 *
Self-image	2.75 ± 0.11	3.88 ± 0.13 ***	3.89 ± 0.15 ***
Mental health	3.73 ± 0.18	4.62 ± 0.07 ***	4.31 ± 0.12 *
Sub-total	3.69 ± 0.10	4.48 ± 0.05 ***	4.31 ± 0.09 ***
Satisfaction	N/A	4.08 ± 0.15	4.02 ± 0.17
Total	N/A	4.44 ± 0.05	4.29 ± 0.09

All values are expressed as the mean ± standard error; SRS-22r: Scoliosis Research Society-22 patient questionnaire, N/A: not applicable; * *p* < 0.05 vs. before surgery, ** *p* < 0.01 vs. before surgery, *** *p* < 0.001 vs. before surgery.

**Table 4 jcm-09-04002-t004:** Comparison of JOABPEQ scores with reference values for low back pain patients.

JOABPEQ Domain	Present Series	Reference Value [18]
Pain-related disorder	86.8 ± 4.2	42.9
Lumbar spine dysfunction	94.0 ± 2.4	58.3
Gait disturbance	95.8 ± 2.7	50.0
Social life disturbance	84.4 ± 3.6	51.4
Psychological disorder	72.9 ± 3.4	47.6

All values are expressed as the mean ± standard error; JOABPEQ: Japanese Orthopaedic Association Back Pain Evaluation Questionnaire.

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
