# Peer review of "Long-Term Surgical Results of Skip Pedicle Screw Fixation for Patients with Adolescent Idiopathic Scoliosis: A Minimum-Ten-Year Follow-Up Study"

_jcm, 2020, doi:10.3390/jcm9124002_

Round 1
Reviewer 1 Report
This paper is an interesting study that provides relevant information for these patients, however, there are some questions that must be resolved.
The beginning of the sentence in the abstract should not be "although ...." it should start more strongly with the clinical context and current evidence.
In the introduction part, they should avoid the terms "we" and generalize in impersonal based on previously published evidence. It would also be interesting to provide a more solid justification for explaining the negative consequences or adverse implications of these interventions.
The last paragraph of the introduction should be the aim of the study
In the material and methods section, it is necessary to provide the necessary information for the study to be republished, and in this case, it is very scarce.No information is provided on how the recruitment was done and where the study was conducted. The data of the approval of the ethics committee that authorized it is not provided, as well as the registration in a clinical database such as clinicaltrials.gov that should have been registered previously. It is also necessary to provide information on the selection criteria, inclusion, and exclusion of the subjects, as well as the characteristics of the physicians who carried it out.
In the cobb angle variable, it is necessary to provide the reference of the reference research study.
The blinding criterion that was carried out is not specified either.
Regarding the results, the Cobb angle not present before surgery calls into question the effectiveness of the technique as there is no comparative data
In the discussion part, it is not possible to speak of improvement of the cobb angle because no values ​​are shown before surgery.
The limitation section should be extended more widely since the level of blinding of the assessors has not been shown and there are many aspects of the methodology to be resolved.
The conclusion cannot be so consistent, it is necessary to assess the type of study that in light of the exposed methods does not allow to affirm this conclusion.
Author Response
Response to Reviewer #1 Comments:
This paper is an interesting study that provides relevant information for these patients, however, there are some questions that must be resolved.
The beginning of the sentence in the abstract should not be "although ...." it should start more strongly with the clinical context and current evidence.
→Thank you for this comment. We have changed the first sentence of the Abstract as follows:
Sentence 1, Paragraph 1, Abstract:
“Skip pedicle screw fixation for adolescent idiopathic scoliosis (AIS) requires fewer screws and can reduce the risk of neurovascular injury as compared with segmental pedicle screw fixation.”
In the introduction part, they should avoid the terms "we" and generalize in impersonal based on previously published evidence. It would also be interesting to provide a more solid justification for explaining the negative consequences or adverse implications of these interventions.
→Thank you for this advice. We have changed the description “To avoid those serious complications, we routinely employ skip pedicle screw fixation for the AIS deformity since it requires fewer screws (Figure 1) [6-9]. Furthermore, skip pedicle fixation also reduces the cost of screws. We earlier described the 2-year results of computer-assisted skip pedicle screw fixation in 62 consecutive cases of AIS containing all Lenke classification types apart from 5C [9]” to the following in the Introduction:
Sentence 4, Paragraph 1, Introduction:
“To avoid those serious complications, Takahashi et al. employ skip pedicle screw fixation for the AIS deformity since it requires fewer screws (Figure 1) and can reduce surgical costs [6-9]. Uehara et al. earlier described the 2-year results of computer-assisted skip pedicle screw fixation in 62 consecutive cases of AIS containing all Lenke classification types apart from 5C [9].”
The last paragraph of the introduction should be the aim of the study
→We appreciate this comment and have added the following last paragraph in the Introduction:
Sentence 1, Paragraph 2, Introduction:
“The purpose of this study is to examine if the corrective position retention and good clinical outcomes of skip pedicle screw fixation are maintained even at 10 years after surgery. To further validate the usefulness of skip pedicle screw fixation for AIS, we examined the radiological and clinical findings of patients after a long observation period of at least 10 years.”
In the material and methods section, it is necessary to provide the necessary information for the study to be republished, and in this case, it is very scarce. No information is provided on how the recruitment was done and where the study was conducted. The data of the approval of the ethics committee that authorized it is not provided, as well as the registration in a clinical database such as clinicaltrials.gov that should have been registered previously.
→Thank you for raising these important points. The study was conducted at our institution. We earlier described the 2-year results of skip pedicle screw fixation in 62 consecutive cases of AIS in a prior study. Among 30 consecutive patients receiving preoperative CT-based navigation-assisted skip pedicle screw fixation for AIS at our hospital between August 2005 and March 2010, 25 who were followed for a minimum of 10 years were retrospectively reviewed. Prior approval of the study was obtained from the investigational review board of our hospital (No. 3500). The complete database of our cohort can be accessed at the Zenodo repository at www.doi.org/10.5281/zenodo.4295859. We have included the following descriptions in the Methods:
Sentence 1, Paragraph 1, Patients, Methods:
“This study was conducted at our institution. We earlier described the 2-year results of skip pedicle screw fixation in 62 consecutive cases of AIS [9]. Among 30 consecutive patients receiving preoperative computed tomography (CT)-based navigation-assisted skip pedicle screw fixation for AIS at our hospital between August 2005 and March 2010, 25 patients (two male and 23 female; mean ± standard deviation [SD] age: 14.4 ± 2.1 years, range: 12–19 years) who were followed for a minimum of 10 years (follow-up rate: 83.3%) were retrospectively reviewed.”
Sentence 9, Paragraph 1, Patients, Methods:
“Prior approval of the study was obtained from the investigational review board of our hospital (No. 3500). Written informed consent for publication was obtained from all patients prior to this study. No financial support or equivalent was received for this investigation. The complete database of the cohort can be accessed at the Zenodo repository (doi.org/10.5281/zenodo.4295859).”
It is also necessary to provide information on the selection criteria, inclusion, and exclusion of the subjects, as well as the characteristics of the physicians who carried it out.
→We agree with the Reviewer and have added the following descriptions in the Methods:
Sentence 4, Paragraph 1, Patients, Methods:
“The inclusion criteria were patients who underwent AIS surgery and follow-up at our hospital. The exclusion criteria were follow-up of less than 10 years and Lenke type 5 curves.”
Sentence 3, Paragraph 1, Patients, Methods:
“All surgeries were performed by one senior spine surgeon specializing in scoliosis (J.T.).”
In the cobb angle variable, it is necessary to provide the reference of the reference research study.
→We have added the following reference [11] as indicated:
“11. Cobb, J.R. Outline for the study of scoliosis. Am Acad Orthop Surg Inst Course Lect 1948, 5, 261-75.”
The blinding criterion that was carried out is not specified either.
→We apologize for this oversight. The first author (M.U.), who was not associated with any surgery in the cohort, performed the evaluation. We have added this point in the Methods:
Sentence 1, Paragraph 1, Evaluation and statistical analysis, Methods:
“The first author (M.U.), who was not associated with any surgery in the cohort, performed the evaluation.”
Regarding the results, the Cobb angle not present before surgery calls into question the effectiveness of the technique as there is no comparative data
In the discussion part, it is not possible to speak of improvement of the cobb angle because no values ​​are shown before surgery.
→Thank you for this important comment. We have added the Cobb angle before surgery in the Results:
Sentence 2, Paragraph 1, Results:
“The preoperative Cobb angle of the main thoracic curve was 59.4°…”
The limitation section should be extended more widely since the level of blinding of the assessors has not been shown and there are many aspects of the methodology to be resolved.
→Thank you for this comment. The first author (M.U.), who was not associated with any surgery in the cohort, performed the evaluation. We agree that no blinding of the assessor was a remaining limitation of this study and have added the following description in the Discussion:
Sentence 2, Paragraph 10, Discussion:
“No blinding of the assessor was also a limitation of this study.”
The conclusion cannot be so consistent, it is necessary to assess the type of study that in light of the exposed methods does not allow to affirm this conclusion.
→We appreciate this comment and have changed the conclusion as follows:
Sentence 1, Paragraph 1, Conclusion:
“Skip pedicle screw fixation for AIS provided significant and sustained radiological and clinical improvements at 10 years after surgery without serious complications.”
Reviewer 2 Report
Interesting study demonstrating that the "Skip pedicles screw " location is acceptable for surgical treatment of AIS especially with sufficient stability for semi and long trem results , not only for oàrthopedic results but also for SRS 22 quality of life scores .Nevertheless the authors concentrate their choice toward the reduction of risk of neuro/vascular injuries. But said very little about the choice of the levels of the insertion of the pedicle screws, according for exemple the levels of the junctional zone in between structural and compensatory ,how was made and what was the choice of LIV? What was the levels and what was the instrumentation of apical vertebrae on convex and what about the concave side ?. We have the impression that the choice of the skip was hazardous and not really pre-determined.This is a very important point to transmit to the spine surgeon their sound philosophy .
Author Response
Response to Reviewer 2 Comments:
Interesting study demonstrating that the "Skip pedicles screw " location is acceptable for surgical treatment of AIS especially with sufficient stability for semi and long term results , not only for orthopedic results but also for SRS 22 quality of life scores .Nevertheless the authors concentrate their choice toward the reduction of risk of neuro/vascular injuries. But said very little about the choice of the levels of the insertion of the pedicle screws, according for example the levels of the junctional zone in between structural and compensatory, how was made and what was the choice of LIV?
→Thank you for your raising these points. Screw insertion was skipped near the apical vertebrae of each curve and in the junctional zone between structural and compensatory curves that were not close to fused end vertebrae. The lower instrumented vertebra (LIV) was determined as the vertebra that last touched the central sacral vertical line when the lumbar modifier was Lenke 1A or 2A. For Lenke 1B and 1C curves, the LIV was selected as the stable vertebra or one vertebra distal to the stable vertebra. For Lenke 3C, 4B, 4C, and 6C curves, we routinely decided the LIV as L3. We have added the following descriptions in the Methods to clarify this:
Sentence 3, Paragraph 1, Surgical technique, Methods:
“Screw insertion was skipped near the apical vertebrae of each curve and in the junctional zone between structural and compensatory curves that were not close to fused end vertebrae.”
Sentence 12, Paragraph 1, Surgical technique, Methods:
“The lower instrumented vertebra (LIV) was decided as the vertebra which last touched the central sacral vertical line when the lumbar modifier was Lenke 1A or 2A. For Lenke 1B and 1C curves, the LIV was determined as the stable vertebra or one vertebra distal to the stable vertebra. For Lenke 3C, 4B, 4C, and 6C curves, we routinely set the LIV as L3.”
What was the levels and what was the instrumentation of apical vertebrae on convex and what about the concave side? We have the impression that the choice of the skip was hazardous and not really pre-determined. This is a very important point to transmit to the spine surgeon their sound philosophy.
→We appreciate this important comment. Pedicle screws were basically inserted near the concave apical vertebrae. For cases in which a 4.5 mm diameter pedicle screw appeared difficult to insert in navigation planning, tape was used instead. However, in cases where only one vertebra was skipped, tape was not used. Since pedicle diameter is wider near the convex vertebrae, we routinely used pedicle screws and skipped one vertebra per inserted vertebra. We have added the following description in the Methods:
Sentence 5, Paragraph 1, Surgical technique, Methods:
“Pedicle screws were routinely inserted near the concave apical vertebrae. For cases in which a 4.5 mm diameter pedicle screw appeared difficult to insert in navigation planning, tape was used instead. However, in cases where only one vertebra was skipped, no tape was used. Since pedicle diameter is wider near the convex vertebrae, we basically used pedicle screws and skipped one vertebra per inserted vertebra.”
Round 2
Reviewer 1 Report
The manuscript has improved and the questions have been answered properly. I congratulate the authors for the research.
Author Response
The manuscript has improved and the questions have been answered properly. I congratulate the authors for the research.
→Thank you for your encouraging comment.
Reviewer 2 Report
good changes according comments of the reviewers but in the discussion the authors have to recognize in their discussion that CT scan of all vertbrae of the length of the fusion to check pecicles anatomy and size is quite a potential damage for long term risk of radiation.
Author Response
good changes according comments of the reviewers but in the discussion the authors have to recognize in their discussion that CT scan of all vertebrae of the length of the fusion to check pedicles anatomy and size is quite a potential damage for long term risk of radiation.
→We agree with this comment and have added the following description in the Discussion:
Sentence 11, Paragraph 2, Discussion:
“Although preoperative CT-based navigation is useful for safely inserting pedicle screws, performing a CT scan of all vertebrae for the length of the fusion to check pedicle anatomy and size carries the potential risk of long-term damage from radiation.”